# Improving Nutrient Use Efficiency of Rice Under Alternative Wetting and Drying Irrigation Combined with Slow-Release Nitrogen Fertilization

**DOI:** 10.3390/plants14101530

**Published:** 2025-05-20

**Authors:** Boyun Lee, Minji Kim, Kyoung Rok Geem, Jwakyung Sung

**Affiliations:** Department of Crop Science, College of Agriculture, Life and Environment Sciences, Chungbuk National University, Cheongju 28644, Republic of Korea

**Keywords:** alternate wetting and drying (AWD), nitrogen metabolism, nitrogen use efficiency (NUE), rice (*Oryza sativa* L.), slow-release fertilizer (SRF)

## Abstract

Rice (*Oryza sativa* L.), a key global staple crop; requires optimized nitrogen (N) and water management to achieve sustainable production under water-limited conditions while minimizing environmental pollution. Improving nitrogen use efficiency (NUE) under limited water availability is essential for sustainable rice production. This study investigated the combined effects of alternate wetting and drying (AWD) water management and slow-release fertilizer (SRF) on NUE photosynthesis; and growth in two rice cultivars; Samgwang (SG) and Milyang#360 (ML). Growth traits; including shoot and grain biomass; were significantly improved under AWD; especially when combined with SRF in the SG cultivar. Photosynthetic rate (Pn) was highest in SG under SRF + AWD treatment. Gene expression analysis revealed that AWD and SRF modulate the expression of nitrogen uptake and assimilation-related genes in a genotype-specific manner. The total nitrogen (N) content; NUE; and nitrogen uptake efficiency (NUpE) were highest under the SRF + AWD treatment. Additionally; the SRF + AWD treatment promoted carbohydrate accumulation in roots; potentially enhancing nutrient uptake under water-limited conditions. These findings highlight the combined application of SRF + AWD as a synergistic and genotype-responsive strategy that improves NUE and crop yield while conserving water and nitrogen resources. Our study provides a practical basis for integrating water and nitrogen management to improve resource efficiency and sustainability in rice cultivation

## 1. Introduction

Rice (*Oryza sativa* L.) is one of the most important cereal crops worldwide and accounts for one-third of global freshwater use during the entire growing season [1], accounting for a water requirement of approximately 2500 L to produce one kilogram of rice (dry weight) [2]. Therefore, considering the shortage of irrigation water resources caused by unexpected environmental impacts, water-saving management is one of the major concerns for sustainable rice production [3]. Furthermore, frequent runoff and leaching during the heavy rainy season are the main factors accounting for up to 56% of nitrogen loss during rice cultivation [4].

Alternate wetting and drying (AWD) is a highly effective strategy not only for saving irrigation water but also for minimizing nutrient losses, including nitrogen. In general, AWD is a water management practice that maintains the water level at 15 to 20 cm below the soil surface [5], ultimately improving water use efficiency (WUE) [6]. Owing to its high effectiveness, AWD has been widely applied in rice production across East Asian countries, including China, the Philippines, and Thailand [7]. In addition to saving water, AWD has also been shown to significantly reduce methane emissions, which typically occur under anaerobic conditions induced by continuous flooding [6].

Nitrogen (N) is an essential nutrient for plants, and crop species including rice heavily depend on it to ensure stable yield and quality. To achieve higher yields, better quality, and improved nitrogen use efficiency in rice production, various nitrogen management practices—such as precise quantitative fertilization—have been implemented [8], Site-specific N management [4] and integrated soil crop system management [9] have been developed with optimized rate and time of N input. Despite various efforts to improve N availability in crop plants, the conventional practice of applying N fertilizer through three or four split broadcast applications remains dominant, often resulting in substantial N losses [10]. Slow- or controlled-release nitrogen fertilizers (S/CRF) have the characteristic of continuously supplying nitrogen to crop plants throughout the entire growth period [11], and they can also reduce ammonia volatilization [12,13] and greenhouse gas emissions such as methane and nitrous oxide [6,14]. Several studies demonstrated that S/CRFs increased the activity of nitrate reductase and glutamine synthetase with improved root architecture due to a continual release of N [15,16,17]. Therefore, S/CRFs can lead to enhanced rice yield and improved N use efficiency [13,18]. Although AWD and SRF have each been understood as effective practices for improving water and nitrogen use efficiency in rice cultivation, few studies have investigated their combined application [6,11]. Fluctuations in soil moisture associated with AWD may affect the solubility and diffusion of nitrogen from SRF, potentially improving the synchronization between nitrogen release and crop nitrogen uptake [19,20]. However, AWD alone can increase the risk of nitrogen losses through leaching and denitrification due to repeated wetting and drying cycles, and may also induce water stress in sensitive rice [21]. In contrast, the nutrient release pattern of SRF is highly dependent on environmental conditions such as temperature and soil moisture, and may not align well with plant nitrogen demand due to its fixed release rate [22]. Therefore, the integration of AWD and SRF may offer complementary benefits that overcome the limitations of each practice, although the physiological mechanisms underlying their interaction remain largely unexplored.

Despite the effectiveness of AWD and SRF in rice production systems, little is known about the combined effects of AWD + SRF on carbon–nitrogen (C–N) metabolism and nitrogen use efficiency (NUE) in rice. To explore the physiological effects of the AWD + SRF combination, the present study employed two water management strategies (CF and AWD) and two nitrogen fertilizer treatments: urea and SRF. In particular, we investigated how soil water variation interacts with nitrogen release patterns to regulate nitrogen assimilation and carbon distribution in rice. The results of this study may provide a better understanding of C–N metabolism under AWD + SRF conditions, offering novel insights into the synchronization of water and nutrient management. These findings could contribute to the identification of genotype-specific strategies and field practices for improving NUE and rice yield, while enhancing the sustainability of rice production systems under climate variability.

## 2. Results

### 2.1. Growth Responses

Growth parameters of rice plants grown under different N and water managements were measured at tillering, heading and harvest time points (Figure 1 and Table 1). At the tillering stage, plant height and tiller number were significantly greater under continuous flooding (CF; 62.0–67.7 cm) compared to alternate wetting and drying (AWD; 57.8–62.2 cm), and the difference between nitrogen sources was more pronounced under AWD conditions. By contrast, at the heading stage, these parameters tended to be higher under AWD, although the differences were not statistically significant. At the harvest stage, shoot and grain biomass per plant were significantly greater under AWD (shoot: 14.2–19.7 g/plant; grain: 4.1–5.6 g/plant) compared to CF (shoot: 11.5–17.4 g/plant; grain: 3.2–5.4 g/plant). Urea, as a nitrogen source, clearly contributed to increased biomass production. No significant differences in growth parameters were observed between the two rice cultivars, SG and ML.

### 2.2. Photosynthesis and Soluble Carbohydrates

Photosynthetic parameters, including the CO_2_ fixation rate, were measured at the heading stage (Table 2). CO_2_ fixation rates ranged from 10.9 to 12.8 µmol m^−2^ s^−1^ under UREA and from 9.5 to 13.1 µmol m^−2^ s^−1^ under SRF. In terms of water management, rates ranged from 9.5 to 12.8 µmol m^−2^ s^−1^ under CF and from 10.9 to 13.1 µmol m^−2^ s^−1^ under AWD. The highest CO_2_ fixation rate was observed in the SRF–AWD treatment of the SG variety, reaching 13.1 µmol m^−2^ s^−1^. In the ML variety, the highest rate was recorded under the UREA + CF treatment, with values ranging from 12.8 to 13.1 µmol m^−2^ s^−1^. The difference between varieties was not significant. Instantaneous carboxylation rates (CEi, calculated as CO_2_ fixation rate per intracellular CO_2_ concentration) in the SG variety did not differ significantly among treatments, whereas in the ML variety, the UREA treatment showed noticeably higher values compared to SRF. Water use efficiency (WUEi, calculated as CO_2_ fixation per unit of evapotranspiration) was significantly higher under the SRF + AWD treatment in the SG cultivar, reaching 3.4 µmol CO_2_ mol^−1^ H_2_O, compared to other treatments (range: 1.3 ± 0.1 to 3.0 ± 0.1 µmol CO_2_ mol^−1^ H_2_O). In contrast, all treatment groups in the ML variety exhibited a similar range of WUEi values, from 2.0 to 3.0 µmol CO_2_ mol^−1^ H_2_O. Total soluble sugar and starch contents were measured in leaf blades and roots at the heading stage (Figure 2). The abundance of soluble sugars in leaf blades was significantly affected by water management rather than nitrogen fertilization in both varieties, with higher levels observed under continuous flooding (CF). The level of soluble sugars in roots was clearly dependent on the combination of treatment factors, showing significantly higher concentrations under CF + UREA and AWD + SRF treatments. Starch in leaf blades was noticeably abundant CF treatment, and SG showed higher accumulation compared to ML. By contrast, the AWD treatment combined with SRF resulted in a remarkable accumulation of starch.

### 2.3. Nitrogen Metabolism

The relative expression levels of genes involved in nitrogen uptake and assimilation were analyzed in leaf blades and roots of both rice varieties under different water and nitrogen treatments (Figure 3). Nitrogen uptake was distinctly mediated by ammonium transporters (AMTs, NH_4_^+^) and nitrate transporters (NRTs, NO_3_^−^). The expression levels of *OsAMT1.1* (high-affinity) and *OsAMT2.1* (low-affinity) did not differ significantly among the treatment groups. By contrast, the high-affinity *OsNRT2.1* gene was markedly upregulated under CF conditions, whereas the low-affinity *OsNRT1.1b* gene was noticeably upregulated under AWD. Neither N form nor variety affected the expression of those genes. In roots, the relative expression levels of *OsNRT2* and *OsGOGAT1* differed markedly between varieties, with higher expression observed in SG under CF and in ML under SRF. *OsNIR*, *OsGS1.1*, and *OsAAT1* were significantly upregulated under CF treatment. The expression of *OsAS1* showed a marked difference in ML, with upregulation observed under UREA + AWD and downregulation under SRF + AWD. In the leaf blades, *OsNIR* and *OsAS1* were remarkably upregulated under CF treatment. The relative expression of *OsGS1.1* was higher in SG under CF and in ML under AWD, respectively, while *OsAAT*1 showed increased expression in UREA + CF and SRF + AWD treatments.

### 2.4. Nitrogen Use Efficiency

Nitrogen use efficiency (NUE) under different water management and nitrogen fertilization treatments was evaluated and compared between the two rice varieties (Table 3). Total nitrogen contents in grains and shoots differed between the two cultivars (SG grain: 1.00 ± 0.02% to 1.09 ± 0.03%; shoot: 0.83 ± 0.05% to 1.02 ± 0.03%; ML grain: 0.89 ± 0.01% to 1.00 ± 0.04%; shoot: 0.83 ± 0.03% to 0.87 ± 0.01%), with consistently higher levels observed in SG. In the SG cultivar, total nitrogen content was highest under the SRF + AWD treatment among all variety treatment combinations, with 1.09 ± 0.03% in the grain and 1.02 ± 0.03% in the shoot (Table 3). In the ML cultivar, total nitrogen content was highest under the UREA + CF treatment in the grain (1.00 ± 0.04%) and under the SRF + CF treatment in the shoot (0.87 ± 0.01%). However, the differences in shoot nitrogen content were not statistically significant (Table 3). Nitrogen uptake efficiency (NUpE) in both cultivars (SG and ML) was more strongly influenced by water management than by the type of nitrogen fertilizer applied (Table 3). Nitrogen utilization efficiency (NUtE) in the SG cultivar was lowest under the SRF + AWD treatment (95.3 ± 0.8 g g^−1^) and highest under the SRF + CF treatment (112.9 ± 2.5 g g^−1^). In contrast, the ML cultivar showed no significant differences in NUtE among treatments. he NUtE of ML was consistently higher than that of SG under all treatment conditions, indicating a greater capacity for nitrogen utilization in ML. In the case of grain nitrogen use efficiency (gNUE), the highest values were observed under the SRF + AWD treatment in the SG cultivar, compared to other treatments. In contrast, no significant differences in gNUE were found among treatments in the ML cultivar (Table 3). The nitrogen harvest index (NHI) showed similar values across all treatments and cultivars, with the highest value observed under the SRF + CF treatment (44.2 ± 5.3%) (Table 3). In the SG cultivar, NUE was highest under the SRF + AWD treatment (103.7 ± 10.2 g g^−1^), whereas in the ML cultivar, the highest NUE was observed under the UREA + AWD treatment (99.5 ± 5.8 g g^−1^) (Table 3). These results suggest that AWD generally promotes higher NUE compared to CF in both cultivars.

### 2.5. Principal Component Analysis (PCA)

Principal component analysis (PCA) was used to evaluate the variations between plant varieties at the metabolome level [23]. To investigate the combined effects of nitrogen fertilizer types (UREA, SRF) and water management methods (CF, AWD) at the heading stage, PCA was performed separately for SG and ML cultivars, and the results are presented in Figure 4. In SG, principal components PC1 and PC2 accounted for 53.6% and 32.9% of the total variance, respectively, explaining a combined 86.5%. In ML, PC1 and PC2 explained 49.9% and 38.8% of the variance, respectively, with a total of 88.7% explained. In both cultivars, AWD was associated with enhanced nitrogen use efficiency (NUE), total dry weight (DW), nitrogen uptake efficiency (NUpE), and total nitrogen content (gTN), whereas CF was associated with higher nitrogen harvest index (NHI), nitrogen utilization efficiency (NUtE), and grain-specific NUE (gNUE).

## 3. Discussion

Improving nitrogen use efficiency (NUE) under limited water availability is a critical challenge for sustainable rice production, especially in the context of increasing water scarcity and nitrogen losses through denitrification, volatilization, and runoff [3,4]. In this study, the combined effects of water management (CF and AWD) and nitrogen fertilizer type (UREA and SRF) were evaluated on growth, physiological traits, and nitrogen metabolism in two rice cultivars, Samgwang (SG) and Milyang#360 (ML).

### 3.1. Growth and Yield Responses to Water and Nitrogen Management

The growth and development of both cultivars were influenced by water management and the type of nitrogen fertilizer (Figure 1 and Table 1), indicating that combination of water management and fertilizers important for rice cultivation. CF promoted plant height and tiller number compared to AWD treatment at tiller and heading stages (Table 1). However, shoot and grain dry weights at the harvest stage were tend to higher under AWD (Table 1). These findings are consistent with previous results showing that AWD improves root development and soil chemical properties, thereby enhancing nutrient uptake and biomass production [24,25]. In particular, UREA + AWD resulted in the highest shoot biomass, while SRF + AWD led to the greatest grain biomass at the harvest stage (Figure 1 and Table 1), suggesting that AWD may be an effective strategy for enhancing biomass production regardless of the nitrogen fertilizer type.

### 3.2. Improved Photosynthetic Performance Under SRF + AWD

Furthermore, we examined rice growth and yield in response to different fertilizer treatments combined with water management. Previous studies have reported that nitrogen fertilizer efficiency can vary depending on water management practices [26]. Therefore, plant growth and physiological responses were evaluated under combinations of UREA and SRF treatments with CF and AWD conditions. The photosynthetic rate is widely regarded as one of the most reliable physiological indicators of crop growth and productivity [27]. The photosynthetic rate is positively associated with NUE, as nitrogen is a key component of photosynthetic enzymes, including RuBisCO. Enhanced photosynthesis, in turn, improves nitrogen assimilation and biomass production efficiency. In this study, the SRF + AWD treatment improved the photosynthetic rate (Pn) and water use efficiency (WUEi) in SG, accompanied by an increase in stomatal conductance (gs) (Table 2). These results are consistent with previous reports suggesting that AWD enhances photosynthetic activity [28], SRF is characterized by its ability to gradually release nitrogen throughout the crop growth period, thereby helping to minimize nitrogen losses such as ammonia volatilization. [29,30]. These results indicate that the SRF + AWD treatment is a highly effective combination for SG, as the sustained and adequate nitrogen supply from SRF contributes maintain sufficient nitrogen availability for photosynthesis under AWD conditions.

### 3.3. Carbohydrate Accumulation and Nitrogen Uptake Efficiency

Nitrogen uptake is regulated by various environmental factors such as soil nutrients and water availability [31,32]. Interestingly, our results confirmed that total nitrogen content in both grain and shoot, along with NUE, NUpE, gNUE, and NHI, were highest under the SRF + AWD treatment in SG (Table 3). Moreover, NUE was consistently higher under AWD treatment, regardless of the type of nitrogen fertilizer applied (Table 3). The effect of SRF was more pronounced in the SG cultivar, as evidenced by the highest total nitrogen content and NUE under the SRF + AWD treatment. In contrast, the ML cultivar showed only a limited response compared to the UREA treatment (Table 3). In our results, the SRF + AWD treatment induced the accumulation of soluble sugars and starch in the roots (Figure 2). Previous studies have demonstrated that carbohydrate metabolism is closely linked to nitrogen uptake and assimilation, playing a key role in the regulation of NUE [33,34]. The increased accumulation of carbohydrates under SRF + AWD treatment suggests an adaptive response to water-saving conditions and a potential improvement in nitrogen uptake [33,35,36,37]. This accumulation of carbohydrates may contribute to osmotic regulation and help sustain root function for efficient nutrient absorption [38].

### 3.4. Gene Expression and Metabolic Profiling Reveal Cultivar-Specific NUE Responses

Furthermore, we investigated the expression of nitrogen uptake- and assimilation-related genes in the leaf blades and roots of both rice cultivars under different nitrogen and water management conditions (Figure 3). The expression patterns of these genes revealed genotype-specific responses to water and nitrogen treatments. Nitrogen transporters are generally classified into high-affinity transport systems (HATS) and low-affinity transport systems (LATS), which function differentially depending on soil nitrogen availability [39]. We analyzed two categories of nitrogen transporters to evaluate their responses to different combinations of nitrogen and water treatments in both cultivars. *OsAMT1.1* and *OsAMT2.1* showed no significant variation across treatments, whereas *OsNRT2.1* was upregulated under CF and *OsNRT1.1b* under AWD, indicating that nitrate uptake is differentially regulated by water management. In roots, the expression of *OsNRT2* and *OsGOGAT1* varied between cultivars—being higher in SG under CF and in ML under SRF. Nitrogen assimilation-related genes such as *OsNIR, OsGS1.1*, and *OsAAT1* were generally upregulated under CF, whereas *OsAS1* showed strong induction in ML under both UREA + CF and SRF + AWD treatments. In leaf blades, *OsNIR* and *OsAS1* were also enhanced by CF, while OsGS1.1 and *OsAAT1* exhibited expression patterns dependent on both genotype and treatment. In addition, PCA analysis revealed that the two cultivars, SG and ML, exhibited distinct metabolomic responses depending on the type of nitrogen fertilizer and water management, each showing significant associations with different nitrogen use-related parameters (Figure 3). These patterns align with physiological traits and suggest a coordinated regulation of nitrogen metabolism, depending on cultivar and environmental conditions.

Taken together, our study demonstrates that the combination of SRF + AWD enhances nitrogen uptake, assimilation, and nitrogen use efficiency (NUE) without compromising biomass production or grain yield, particularly in the SG cultivar. Conversely, the ML cultivar exhibited a relatively stable response regardless of the nitrogen fertilizer type. Compared to SG, the ML cultivar showed weaker responses to SRF + AWD, likely due to its limited root adaptability, reduced carbohydrate accumulation, and less responsive nitrogen metabolism. Under SRF + AWD, ML exhibited lower root sugar and starch accumulation, potentially limiting nitrogen uptake and transport under water-saving conditions [34,35,37]. Moreover, nitrogen assimilation genes such as *OsGS1.1* and *OsAAT1* were less induced in ML, suggesting lower transcriptional responsiveness to SRF-based nitrogen. Despite consistently high nitrogen utilization efficiency (NUtE), ML’s physiological and molecular inflexibility may have limited further improvements in NUE. These results emphasize the importance of genotype-specific nutrient and irrigation strategies for sustainable rice production.

These results suggest that genotype-specific management strategies are essential for optimizing rice cultivation [40]. Therefore, although CF and UREA treatments were effective in promoting early growth during the tillering and heading stages, the SRF + AWD treatment demonstrated greater advantages in terms of nitrogen use efficiency (NUE) and grain yield over the entire growth period (Table 1 and Table 3). How SRF + AWD enhances NUE is supported by physiological, biochemical, and molecular evidence from our results. Physiologically, the SRF + AWD treatment enhanced photosynthetic rate (Pn), water use efficiency (WUEi), and stomatal conductance (gs), particularly in the SG cultivar (Table 2), supporting improved nitrogen assimilation. Biochemically, elevated levels of soluble sugars and starch were detected in the roots under SRF + AWD conditions (Figure 2), likely providing the energy required to support nitrogen uptake and related metabolic processes. Molecularly, key nitrogen-related genes such as *OsNRT1.1b*, *OsGS1.1*, and *OsAAT1* were more highly expressed in SG under SRF + AWD treatment (Figure 3), indicating enhanced nitrogen transport and assimilation at the transcriptional level. Furthermore, to enhance sustainable rice production under conditions of water scarcity and nutrient loss, long-term studies are needed to evaluate the effectiveness of AWD across a broader range of rice genotypes and nitrogen fertilizer types. Collectively, this study provides that integrated water and nitrogen management strategies optimized for rice genotypes have the potential to enhance nitrogen use efficiency, grain yield, and environmental sustainability.

Further studies are required to validate the agronomic advantages of the SRF + AWD combination through comprehensive evaluations of yield and grain quality. Such efforts will be essential for developing practical, field-applicable management strategies that optimize productivity while improving resource use efficiency and promoting environmental sustainability. Nevertheless, this study was conducted under controlled greenhouse conditions, which may not fully represent open-field environments or the variability encountered across different field conditions. Although soil microbial communities are also present in greenhouse conditions, their composition and functional activity may differ from those in the field due to differences in microbial diversity, soil structure, and environmental interactions. Soil microbes play an important role in regulating nitrogen availability by mediating urea hydrolysis, nitrogen release from SRF, and the transformation of ammonium and nitrate [41]. These microbially mediated processes may influence nitrogen uptake and fertilizer efficiency under field conditions. In addition, environmental factors such as rainfall and temperature fluctuations in the field may further affect nitrogen dynamics and plant responses. Therefore, field-based studies across various environments and genotypes are needed to verify the consistency and applicability of the SRF + AWD treatment under practical conditions.

## 4. Materials and Methods

### 4.1. Plant Materials and Treatments

The experiment was conducted at a greenhouse of Chungbuk National University, Cheonju, Republic of Korea (36°37′49″ N 127°27′05″ E) from May to October, 2023. Seeds of rice (*Oryza sativa* L. var. ‘Samgwang, SG’, ‘Milyang#360, ML’) were germinated for two days in an incubator (25 °C, in darkness), and seedlings were grown in a growth chamber (200 ± 20 μmol m^−2^ s^−1^ photosynthetic photon flux density under 12-h photoperiod, relative humidity of 60% and day/night temperature of 25/20 °C) until reached at the 3rd to 4th-leaf stage. Uniformly growing rice plants were then transplanted into containers (1/20,000 ha) filled with sandy loam soil characterized by a 5.0 of pH, 0.6 dS/m of an electrical conductivity (EC), 0.22 g/kg of soil organic matter (SOM), 74.5 mg kg^−1^ inorganic N, 389 mg kg^−1^ soluble K, 458 mg kg^−1^ soluble Ca, and 131 mg kg^−1^ soluble Mg. Standard fertilization of N (90 kg ha^−1^)-P_2_O_5_ (45 kg ha^−1^)-K_2_O (57 kg ha^−1^) recommended by Rural Development Administration, South Korea.

Nitrogen (N) was fertilized with an equivalent to 90 kg ha^−1^ with urea (46% N) or slow releasing fertilizer (SRF: Danhanbeon, Chobi Co. Ltd., Seoul, Republic of Korea, N-P-K: 18-7-9%), and was split into three doses: 50% applied at basal (before transplanting), 30% at the tillering stage (15 days after transplanting, DAT), and 20% at the panicle initiation stage (60 DAT). P_2_O_5_ (fused superphosphate) was fully applied before transplanting, while K_2_O (potassium chloride) was split, with 70% applied at basal and 30% at the panicle initiation stage. The amount of P_2_O_5_ and K_2_O equivalent to the shortage by SRF was just applied. Water management, continuous flooding (CF) and alternative wetting and drying (AWD), was employed (Appendix A). CF was irrigated with 2–5 cm of water table from surface soil until mid-season drainage, while watering in AWD was supplied as soil surface was cracked. Four treatments composed of UREA + CF, UREA + AWD, SRF + CF and SRF + SRF were arranged in a completely randomized design with three replicates.

### 4.2. Photosynthesis Measurement

Photosynthetic parameters, photosynthetic rate (P*_n_*), stomatal conductance (g*_s_*), intercellular CO_2_ concentration (C_i_) and transpiration rate (E), were measured from the upper fully expanded leaves at the heading stage using a portable photosynthesis system (LC-PRO +, ADC BioScientific Ltd., Hertfordshire, London, UK). Measurements were taken between 10:00~12:00 AM at ambient CO_2_ concentration (400–410 mmol m^−3^) under a light intensity of 1000 ± 200 μmol m^−2^ s^−1^ photosynthetic photon flux density, and an ambient temperature of 30 ± 3 °C. The instantaneous carboxylation efficiency (CE_i_) and water use efficiency (WUE_i_) and were calculated with photosynthetic parameters (Equations (1) and (2)).CE_i_ (µmol m^−2^ s^−1^) (µmol mol^−1^)^−1^ = P_n_/C_i_(1)WUE_i_ (µmol CO_2_ mol^−1^ H_2_O) = P_n_/E(2)

### 4.3. Soluble Sugars and Starch Analysis

The upper fully expanded leaf blades and roots at heading stage were taken to analyze soluble sugars and starch. Powered sample (0.2 g, dry weight) were extracted with 10 mL of 80% ethanol and the supernatant was collected as a fraction of soluble sugars. After evaporated, the residues (starch fraction) were dissolved in distilled water, mixed with 2 volumes of 0.2% anthrone in a concentrated H_2_SO_4_, and carbohydrate content was measured at 630 nm with a spectrophotometer (UV-1900i, SHIMADZU, Kyoto, Japan) [42]. Glucose was used as a standard.

### 4.4. RNA Extraction and Quantitative Real-Time PCR Analysis

Total RNA extraction using TRIzol reagent (Invitrogen, Carlsbad, CA, USA) according to the manufacturer’s instructions was extracted from leaf blades and roots of rice treated with different form and rate of N fertilization at the heading time. The purity and concentration of the extracted RNA were estimated using NanoDrop (Thermo Fisher Scientific, Madison, WI, USA), and checked on 1.0% agarose gel. Total RNA (1 μg) and RT PreMix Kit (iNtRON Biotechnology, Inc., Seongnam, Republic of Korea) with Oligo (dT) primers were used to synthesize first-strand cDNA using the following PCR conditions; 60 min at 45 °C to cDNA synthesis and 5 min at 95 °C to Rtase inactivation step. Quantitative real-time PCR was performed by using a Real-Time PCR machine (CFX Opus 96, Bio-Rad, Hercules, CA, USA) with technical triplicates with the manufacturer’s instructions. The reaction mixture consisted of 1 μL of cDNA template, 2 μL each of 10 mM forward and reverse primer (Appendix A), and 5 μL SYBR Green Q Master mix (Labopass, Cosmo Genetech, Seoul, South Korea). The PCR conditions consisted of pre- denaturation step at 95 °C for 3 min, followed by 40 cycles of denaturation at 95 °C for 15 s, annealing temperature of each primer (Appendix A) for 15 s, and elongation (72 °C, 15 s). This step was followed by a melting curve, ranging from 65 to 95 °C at a heating rate of 0.5 °C/s. A quantification method (2^−∆∆Ct^) was used [43] and the variation in expression was estimated using triplicate for each cDNA sample. An actin gene was used as a reference in the qRT-PCR, and primer sequences used for qRT-PCR were designed by Primer 3 (version 0.4.0) software [38,44] (Appendix A).

### 4.5. Total Nitrogen and Nitrogen Use Efficiency (NUE)

The harvested rice plants were divided into shoots (leaf blade and sheath) and grains, immediately dried at 80 °C for 72 h, and 0.5 g of dried samples used to measure total nitrogen content using Rapid MAX N exceed (Elementar, vario MAX CN, Mount Laurel, Elementar Americas, Ronkonkoma, NY, USA). With total nitrogen content and dry weight, nitrogen use efficiency was assessed as the follows.Nitrogen use efficiency (NUE) = Yield (g)/Applied N (g)(3)Nitrogen uptake efficiency (NUpE) = Acquired N (g)/Applied N (g)(4)Nitrogen utilization efficiency (NUtE) = Yield (g)/Acquired N (g)(5)Nitrogen use efficiency of grain (gNUE) = Grain weight (g)/Applied N (g)(6)Nitrogen harvest index (NHI, %) = Grain N accumulation/Total N accumulation (Grain + Shoot)(7)

### 4.6. Statistical Analysis

Data were analyzed using RStudio (Version 4.1.3) with one-way ANOVA followed by Tukey’s HSD test if *p* < 0.05. *T*-test was used to compare differences between cultivars. Principal component analysis (PCA) was also performed to assess the correlation between NUE variables.

## 5. Conclusions

This study demonstrates that the combination of slow-release fertilizer (SRF) and alternate wetting and drying (AWD) irrigation significantly improves nitrogen uptake, assimilation, and nitrogen use efficiency (NUE) without reducing biomass or grain yield, particularly in the Samgwang (SG) cultivar. In contrast, the Milyang#360 (ML) cultivar exhibited more stable responses with relatively less variation across the treatment groups. Principal component analysis (PCA) further revealed that the two cultivars responded differently at the metabolic level to nitrogen and water management, with SG showing stronger associations with traits such as NUE, total nitrogen content, and nitrogen uptake efficiency. These findings emphasize the importance of developing genotype-specific nutrient and irrigation management strategies. In particular, the SRF + AWD combination was shown to synchronize nitrogen supply and plant uptake effectively, contributing to improved nitrogen metabolism and root carbohydrate accumulation in SG. Meanwhile, the limited physiological flexibility of ML under SRF + AWD highlights the need for genotype-specific management approaches according to varietal characteristics. Altogether, the SRF + AWD combination represents a promising and sustainable strategy to improve nutrient use efficiency and sustain rice productivity under water-limited conditions. These results provide a basis for field-level applications and suggest the need for further multi-genotype evaluations to support broader adoption under diverse environmental conditions.

## Figures and Tables

**Figure 1 plants-14-01530-f001:**
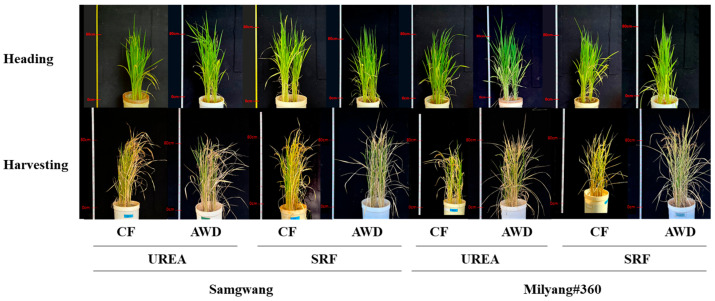
Rice growth at the heading and harvest stag es was evaluated under different nitrogen and water management conditions. The nitrogen source was either urea or slow-release fertilizer (SRF), and water was managed using either continuous flooding (CF) or alternate wetting and drying (AWD).

**Figure 2 plants-14-01530-f002:**
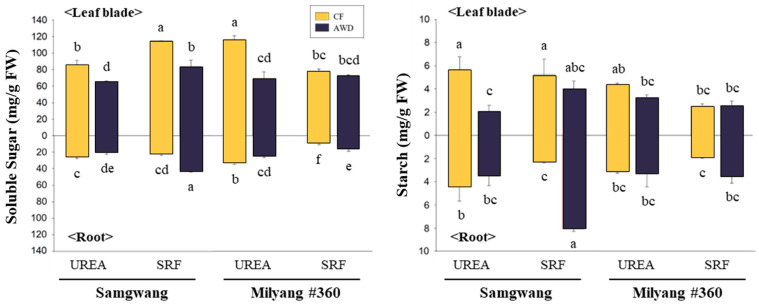
Soluble sugar and starch contents in leaf blades and roots of both rice cultivars at the heading stage under different nitrogen and water management conditions. Nitrogen was supplied either as urea or slow-release fertilizer (SRF), and water was managed using either continuous flooding (CF) or alternate wetting and drying (AWD). Different letters indicate significant differences according to Tukey’s test at *p* < 0.05 (*n* = 3). Differences between cultivars are indicated above and below the graph.

**Figure 3 plants-14-01530-f003:**
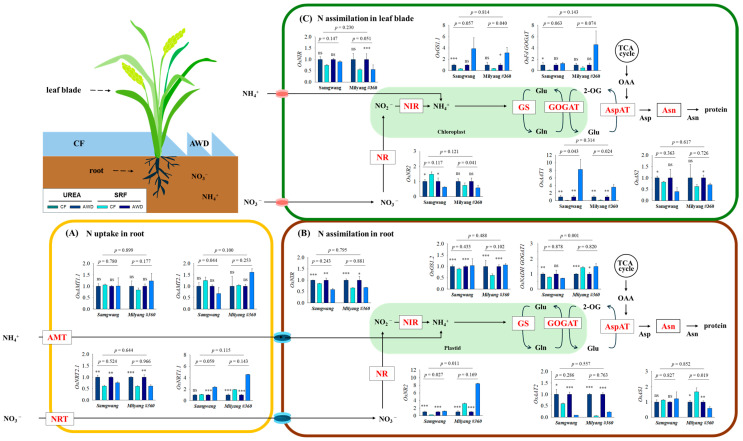
Relative gene expression (log_2_ scale) associated with nitrogen metabolism as measured by qRT-PCR. Nitrogen was supplied either as urea or slow-release fertilizer (SRF), and water was managed using either continuous flooding (CF) or alternate wetting and drying (AWD). Statistical significance based on the *t-*test is indicated as follows: ns (not significant), * (*p* < 0.05), ** (*p* < 0.01), and *** (*p* < 0.001) (*n* = 3). Gene information is provided in Appendix A. AMT: ammonium transporter; NRT: nitrate transporter; NR: nitrate reductase; NIR: nitrite reductase; GS: glutamine synthetase; Fd-GOGAT: ferredoxin-dependent glutamate synthase; NADH-GOGAT: NADH-dependent glutamate synthase; AspAT and AAT: aspartate aminotransferase; Asn and AS: asparagine.

**Figure 4 plants-14-01530-f004:**
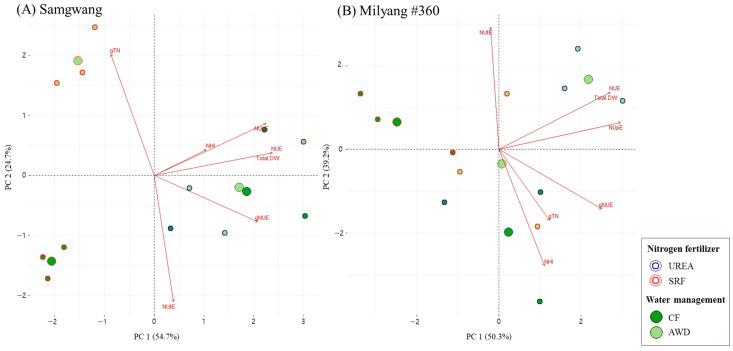
Principal component analysis (PCA) of nitrogen use efficiency parameters in rice plants grown under different nitrogen and water management conditions. Nitrogen was supplied either as urea or slow-release fertilizer (SRF), and water was managed using either continuous flooding (CF) or alternate wetting and drying (AWD).

**Table 1 plants-14-01530-t001:** Growth and biomass production of rice at the tillering, heading, and harvest stages were assessed under different nitrogen and water management conditions. Nitrogen was supplied either as urea or slow-release fertilizer (SRF), and water was managed using either continuous flooding (CF) or alternate wetting and drying (AWD). Statistical differences among nitrogen and water management treatments within each rice variety were analyzed using Tukey’s HSD test at *p* < 0.05 (*n* = 3), and differences between cultivars were evaluated using the student’s *t*-test. Significance levels: * *p* < 0.05; ** *p* < 0.01; *** *p* < 0.001; ns: not significant. Different letters indicate statistically significant differences (*p* ≤ 0.05).

Variety	Nitrogen	Water	Tillering		Heading		Harvest	
			Plant Height (cm)	Tiller (no. Plant^−1^)	Plant Height (cm)	Tiller (no. Plant^−1^)	Shoot DW (g Plant^−1^)	Grain DW (g Plant^−1^)
Samgwang (SG)	UREA	CF	66.5 ± 4.2 a	11 ± 1ab	87.1 ± 5.4 ns	14 ± 1 ns	17.4 ± 3.3 a	5.4 ± 0.4 a
		AWD	58.0 ± 0.5 b	6 ± 1c	94.8 ± 4.5	14 ± 1	18.0 ± 1.6 a	4.7 ± 0.6 ab
	SRF	CF	62.0 ± 1.1 ab	13 ± 1 a	85.7 ± 1.8	14 ± 2	11.5 ± 0.1 b	4.1 ± 0.2 b
		AWD	60.1 ± 2.1 b	8 ± 2 bc	91.0 ± 7.8	14 ± 3	16.3 ± 1.1 ab	5.6 ± 0.2 a
		*f* value	6.6408 (*)	27.333 (***)	1.7723 (ns)	0.3333 (ns)	7.2863 (*)	3.9854 (**)
Milyang#360 (ML)	UREA	CF	67.7 ± 0.5 a	10 ± 1 a	86.0 ± 2.2 ns	14 ± 1 ns	12.5 ± 1.6 b	4.5 ± 1.2 ns
		AWD	57.8 ± 1.9 c	7 ± 0 b	90.8 ± 9.3	14 ± 1	19.7 ± 0.8 a	4.4 ± 0.6
	SRF	CF	66.5 ± 2.2 a	9 ± 2 ab	85.1 ± 5.2	12 ± 4	11.7 ± 0.7 b	3.2 ± 0.7
		AWD	62.2 ± 1.2 b	10 ± 1 ab	85.9 ± 2.6	18 ± 2	14.2 ± 1.7 b	4.1 ± 0.5
		*f* value	24.386 (***)	5.3333 (*)	0.6582 (ns)	2.5476 (ns)	23.791 (***)	1.7597 (ns)
Variety		*p* value	0.5372	0.7586	0.3086	0.6509	0.601	0.09464

**Table 2 plants-14-01530-t002:** Photosynthetic parameters at the heading stage were influenced by different nitrogen and water management conditions. Nitrogen was supplied either as urea or slow-release fertilizer (SRF), and water was managed using either continuous flooding (CF) or alternate wetting and drying (AWD). Statistical differences among nitrogen and water management treatments within each rice variety were analyzed using Tukey’s HSD test at *p* < 0.05 (*n* = 3), and differences between cultivars were evaluated using the student’s *t*-test. Significance levels: * *p* < 0.05; *** *p* < 0.001; ns: not significant. Different letters indicate statistically significant differences (*p* ≤ 0.05).

Variety	Nitrogen	Water	P*_n_* (μmol CO_2_ m^−2^ s^−1^)	g*_s_* (μmol H_2_O m^−2^ s^−1^)	E (mmol H_2_O m^−2^ s^−1^)	C_i_ (μmol CO_2_ m^−2^ s^−1^)	CE_i_ [(μmol m^−2^ s^−1^) (μmol mol^−1^)]^−1^	WUE_i_ (μmol CO_2_ mol^−1^ H_2_O)
Samgwang (SG)	UREA	CF	11.3 ± 1.0 ab	0.4 ± 0.0 b	4.4 ± 1.6 b	316.0 ± 1.0 b	0.04 ± 0.00 ns	2.1 ± 0.2 ab
		AWD	10.9 ± 0.7 ab	0.3 ± 0.0 b	8.5 ± 0.3 a	288.0 ± 1.7 c	0.04 ± 0.00	1.3 ± 0.1 b
	SRF	CF	9.5 ± 1.8 b	0.3 ± 0.0 b	7.8 ± 0.3 a	314.0 ± 1.0 b	0.03 ± 0.01	1.2 ± 0.2 b
		AWD	13.1 ± 0.3 a	0.8 ± 0.1 a	3.8 ± 0.1 b	343.3 ± 1.0 a	0.04 ± 0.00	3.4 ± 0.0 a
		*f* value	5.2992 (*)	27.838 (***)	25.854 (***)	173.58 (***)	3.1141 (ns)	6.5197 (*)
Milyang#360 (ML)	UREA	CF	12.8 ± 1.0 ns	0.5 ± 0.1 b	5.6 ± 1.4 ns	355.0 ± 19.1 a	0.04 ± 0.00 ab	2.4 ± 0.9 ns
		AWD	12.6 ± 0.6	0.3 ± 0.1 c	4.2 ± 0.1	282.7 ± 2.9 b	0.04 ± 0.00 a	3.0 ± 0.1
	SRF	CF	10.6 ± 2.0	0.3 ± 0.0 c	5.9 ± 1.5	305.0 ± 6.2 b	0.03 ± 0.01 b	2.0 ± 1.0
		AWD	11.2 ± 0.2	0.6 ± 0.0 a	3.9 ± 0.0	344.0 ± 4.4 a	0.03 ± 0.00 b	2.8 ± 0.0
		*f* value	2.7221 (ns)	45.656 (***)	2.7117 (ns)	31.684 (***)	6.012 (*)	1.503 (ns)
Variety		*p* value	0.5429	0.8175	0.366	0.7655	0.6671	0.3635

**Table 3 plants-14-01530-t003:** Nitrogen use efficiency of rice cultivars grown under different nitrogen and water management conditions. Nitrogen was supplied either as urea or slow-release fertilizer (SRF), and water was managed using either continuous flooding (CF) or alternate wetting and drying (AWD). Statistical differences among nitrogen and water management treatments within each rice variety were analyzed using Tukey’s HSD test at *p* < 0.05 (*n* = 3), and differences between cultivars were evaluated using the student’s *t*-test. Significance levels: * *p* < 0.05; ** *p* < 0.01; *** *p* < 0.001; ns: not significant. Different letters indicate statistically significant differences (*p* ≤ 0.05).

Cultivar	Nitrogen	Water	Grain T-N (%)	Shoot T-N (%)	NUE (g g ^−1^)	NUpE (g g ^−1^)	NUtE (g g ^−1^)	gNUE (g g ^−1^)	NHI (%)
Samgwang (SG)	Urea	CF	1.00 ± 0.02 b	0.93 ± 0.05 ab	86.3 ± 9.6 ns	0.8 ± 0.1 b	104.4 ± 4.2 a	35.8 ± 2.6 a	43.7 ± 4.1 b
		AWD	1.02 ± 0.03 ab	0.90 ± 0.02 bc	96.6 ± 9.5	0.9 ± 0.1 ab	105.6 ± 2.1 b	36.8 ± 4.6 ab	40.9 ± 2.1 b
	SRF	CF	1.02 ± 0.03 ab	0.80 ± 0.05 c	83.2 ± 1.3	0.7 ± 0.0 b	112.9 ± 2.5 b	32.1 ± 1.7 b	44.4 ± 2.3 a
		AWD	1.09 ± 0.03 a	1.02 ± 0.03 a	103.7 ± 10.2	1.1 ± 0.1 a	95.3 ± 0.8 c	43.3 ± 1.2 a	43.6 ± 3.8 ab
		*f* value	5.9705 (*)	5.066 (**)	3.7165	8.5881 (**)	21.494 (***)	8.9854 (**)	7.6725 (**)
Milyang#360 (ML)	Urea	CF	1.00 ± 0.04 a	0.83 ± 0.01 ns	92.0 ± 9.8 ns	0.8 ± 0.1 ns	112.0 ± 2.4 ns	36.6 ± 7.2 ns	44.2 ± 5.3 a
		AWD	0.96 ± 0.03 ab	0.83 ± 0.02	99.5 ± 5.8	0.9 ± 0.0	114.2 ± 1.1	34.0 ± 5.0	37.2 ± 3.9 b
	SRF	CF	0.89 ± 0.01 b	0.87 ± 0.01	84.4 ± 8.7	0.7 ± 0.1	114.0 ± 0.7	32.4 ± 5.7	38.9 ± 2.9 ab
		AWD	0.97 ± 0.04 ab	0.83 ± 0.02	84.1 ± 1.7	0.7 ± 0.0	113.5 ± 4.2	32.2 ± 3.9	42.1 ± 6.0 ab
		*f* value	5.269 (*)	4.0586 (ns)	3.1028	3.3273	0.4617	1.7597	4.1781 (*)
Variety		*p* value	17.887 (***)	8.1165 (**)	0.349	3.8973	17.796	2.5016	2.2892

## Data Availability

The data presented in this study are available on request from the first author or corresponding author.

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
