# Peer review of "Improving Nutrient Use Efficiency of Rice Under Alternative Wetting and Drying Irrigation Combined with Slow-Release Nitrogen Fertilization"

_plants, 2025, doi:10.3390/plants14101530_

Round 1

Reviewer 1 Report

Comments and Suggestions for Authors

Review on „Improving nutrient use efficiency of rice under alternative wetting and drying irrigation combined with slow release nitrogen fertilization”

The topic of the manuscript fits to the goal and scope of Plants, MDPI. The topic of the manuscript is up-to-data and contains many valuable results.

The study investigated the combined effects of alternate wetting and drying water management and slow-release fertilizer on nitrogen use efficiency, photosynthesis, and growth in two rice cultivars in a greenhouse experiment. The authors measured the photosynthetic efficiency, the soluble sugar and starch content in rice, total nitrogen; and they calculated the nitrogen use efficiency. They concluded that combining slow-release fertilizer with alternate wetting and drying irrigation significantly improves nitrogen use efficiency, nitrogen uptake, and assimilation without reducing yield, especially in the SG rice cultivar. The ML cultivar showed more stable but less responsive results. Principal component analysis showed different metabolic responses between the cultivars, with SG more strongly linked to improved nitrogen traits. These results highlight the need for genotype-specific nutrient and irrigation strategies, and confirm that slow-release fertilizer with alternate wetting and drying is an effective, sustainable method to optimize rice productivity under limited water conditions.

Before publishing, some minor corrections are needed:

Please carefully check the Instructions for the authors and restructure the manuscript. Please check carefully the superscript and subscript in the whole manuscript.

Abstract:

Please provide exact numerical data, not only „increase” or „decrease”.

Keywords:

please arrange the keywords in alphabetical order.

Author Response

We are grateful to the reviewer for the careful evaluation and helpful suggestions, which greatly enhanced the quality of our manuscript. Our detailed responses to each comment are provided in the attached file. In addition, the newly added or significantly revised parts in the manuscript have been highlighted in yellow for easy reference.

Reviewer 2 Report

Comments and Suggestions for Authors

This study evaluates the synergistic effects of alternate wetting and drying (AWD) irrigation and slow-release fertilizer (SRF) on nitrogen use efficiency (NUE), photosynthesis, and growth in rice. The topic holds significant theoretical and practical importance. The experimental design is rigorous, and the data are comprehensive, providing valuable insights for water- and nutrient-efficient rice cultivation. However, certain aspects require clarification or supplementation to enhance the paper's completeness and readability.

1. The introduction could better highlight the novelty of combining AWD and SRF, will water management affect the release of slow-release fertilizers? Have there been any previous studies on this? What are the potential mechanisms? e.g., by contrasting limitations of studies that applied AWD or SRF alone.

2. Line 77 check the unit of SOM

3. Line 81 90 kg ha-1 should be 90 kg ha-1 and also P2O5, please check

  1. Specify SRF composition (e.g., coating material, release rate) to improve reproducibility
  2. Clarify the significance threshold for Tukey’s test (e.g., p<0.05) and ensure consistent labeling of significant differences (e.g., letter notation) in figures/tables.
  3. Ensure consistency between textual descriptions (e.g., "significant") and table annotations (e.g., "ns" for non-significant) in Table 2.
  4. Clarify units for carbohydrate content (e.g., mg/g DW) in Figure 3.
  5. Further explain why the ML cultivar showed weaker responses to SRF+AWD (e.g., root architecture or nitrogen metabolic pathway differences?)
  6. Cite additional literature to support the hypothesis that "carbohydrate accumulation promotes nitrogen uptake" (e.g., studies on root carbon allocation and nitrogen absorption coupling).
  7. Minor grammatical errors need correction (e.g., "The SRF+AWD strategy improve" → "improves" in the abstract).
  8. Standardize reference formatting (e.g., italicized journal abbreviations, author name styles).
  9. The discussion should be organized based on the key research findings. And also should discuss potential limitations of greenhouse conditions (e.g., differences from field environments, such as soil microbiota or rainfall interference)

Author Response

(The authors gave the same response as above.)

Reviewer 3 Report

Comments and Suggestions for Authors

This manuscript conducted the Improving nutrient use efficiency of rice under alternative wetting and drying irrigation combined with slow release nitrogen fertilization. This study was useful and interesting. My major concerns were as followings:

  1. The abstract need to be improved to highlight the theme of MS.
  2. The design need to be further stated, especially the How to quantify alternative wetting and drying irrigation? How about the mechanism of Improving nutrient use efficiency? I suggested to add some physiological and biochemical, molecular experiments to reveal the mechanism and increase the depth of MS.
  3. Discussion section also need to improved, I suggest to add the subtitles to clearly discuss the results with previous study to highlight the theme of MS.
  4. Language could be polished by native English speakers.

Author Response

(The authors gave the same response as above.)

Round 2

Reviewer 2 Report

Comments and Suggestions for Authors

The author has revised and improved the MS in response to the comments from the reviewers, and it is highly recommended that this work be accepted.

Author Response

We sincerely thank the reviewer for taking the time to review our manuscript and for providing constructive feedback. In response, we have thoroughly reviewed the entire manuscript and made revisions accordingly to enhance its overall clarity and quality.

Reviewer 3 Report

Comments and Suggestions for Authors

Authors have addressed all the comments carefully, I agree to accept this manuscript after careful revision throughout this MS.

Author Response

(The authors gave the same response as above.)
